# A Bayesian Approach to Robust Inverse Reinforcement Learning

**Ran Wei[1], Siliang Zeng[2], Chenliang Li[1], Alfredo Garcia[1], Anthony McDonald[3], Mingyi Hong[2]**
[1]Texas A&M University, [2]University of Minnesota, [3]University of Wisconsin
{rw422, chenliangli, alfredo.garcia}@tamu.edu
{zeng0176, mhong}@umn.edu, {admcdonald}@wisc.edu

**Abstract:** We consider a Bayesian approach to offline model-based inverse reinforcement learning (IRL). The proposed framework differs from existing offline model-based IRL approaches by performing *simultaneous* estimation of the expert's reward function and subjective model of environment dynamics. We make use of a class of prior distributions which parameterizes how accurate the expert's model of the environment is to develop efficient algorithms to estimate the expert's reward and subjective dynamics in high-dimensional settings. Our analysis reveals a novel insight that the estimated policy exhibits robust performance when the expert is believed (a priori) to have a highly accurate model of the environment. We verify this observation in the MuJoCo environments and show that our algorithms outperform state-of-the-art offline IRL algorithms.[1]

**Keywords:** Inverse Reinforcement Learning, Bayesian Inference, Robustness

## 1 Introduction

Inverse reinforcement learning (IRL) is the problem of extracting the reward function and policy of a value-maximizing agent from its behavior [1, 2]. IRL is an important tool in domains where manually specifying reward functions or policies is difficult, such as in autonomous driving [3], or when the extracted reward function can reveal novel insights about a target population and be used to device interventions, such as in biology, economics, and human-robot interaction studies [4, 5, 6]. However, wider applications of IRL face two interrelated algorithmic challenges: 1) having access to the target deployment environment or an accurate simulator thereof and 2) robustness of the learned policy and reward function due to the covariate shift between the training and deployment environments or datasets [7, 8, 9].

In this paper, we focus on model-based offline IRL to address challenge 1). A notable class of model-based offline IRL methods estimate the dynamics and reward in a *two-stage* fashion (see Figure 1) [10, 11, 12, 13]. In the first stage, a dynamics model is estimated from the offline dataset. Then, the dynamics model is fixed and used as a simulator to train the reward and policy in the second stage. To overcome covariate shift in

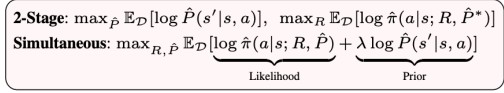

Figure 1: Objectives of the traditional two-stage IRL and the proposed simultaneous estimation approach of Bayesian model-based IRL.

the estimated dynamics, recent methods design density estimation-based "pessimistic" penalties to prevent the learner policy from entering uncertainty regions in the state-action space (i.e., space not covered in the demonstration dataset) [14, 13, 12].

We instead approach IRL from a Bayesian modeling perspective, where we *simultaneously* estimate the expert's reward function and their *internal* model of the environment dynamics. The core idea is that expert decisions convey their beliefs about the environment [15, 16] and thus should affect the update direction of the dynamics model. This Bayesian model of perception and action, related to

---

[1]https://github.com/rw422scarlet/bmirl_tf

7th Conference on Robot Learning (CoRL 2023), Atlanta, USA.

Bayesian Theory of Mind [15], has been used to understand human biases encoded in the internal dynamics in simple and highly constrained domains [17, 16, 18, 19, 20, 21, 22, 23, 24]. In contrast to these works, we study how a Bayesian model enables learning high-performance and robust policies.

We first propose a class of priors parameterizing how accurate we believe the expert's model of the environment is. We then present our *key observation* that if the expert is believed a priori to have a highly accurate model, robustness emerges naturally in the Bayesian modeling framework by a *simultaneous* estimation approach in which planning is performed against the worst-case dynamics outside the offline data distribution. We then analyze how varying the prior affects the performance of the learner agent and pair our analysis with a set of algorithms which extend previous simultaneous estimation approaches [18, 19] to high-dimensional continuous-control settings. We show that the proposed algorithms outperform state-of-the-art (SOTA) offline IRL methods in the MuJoCo environments without the need for designing pessimistic penalties.

## 2 Preliminaries

### 2.1 Markov Decision Process

We consider modeling agent behavior using infinite-horizon *entropy-regularized* Markov decision processes (MDP; [25]) defined by tuple $(\mathcal{S}, \mathcal{A}, P, R, \mu, \gamma)$ with state space $\mathcal{S}$, action space $\mathcal{A}$, transition probability distribution $P(s'|s,a) \in \Delta(\mathcal{S})$, reward function $R(s,a) \in \mathbb{R}$, initial state distribution $\mu(s_0) \in \Delta(\mathcal{S})$, and discount factor $\gamma \in (0,1)$. We denote the discounted occupancy measure as $\rho_P^\pi(s,a) = \mathbb{E}_{\mu,P,\pi}\left[\sum_{t=0}^\infty \gamma^t P(s_t = s, a_t = a)\right]$ and the marginal state-action distribution as $d_P^\pi(s,a) = (1-\gamma)\rho_P^\pi(s,a)$. We further denote the discounted occupancy measure starting from a specific state-action pair $(s,a)$ with $\rho_P^\pi(\tilde{s}, \tilde{a}|s,a)$. The agent selects actions from an optimal policy $\pi(a|s) \in \Delta(\mathcal{A})$ that achieves the maximum expected discounted cumulative rewards and policy entropy $\mathcal{H}(\pi(\cdot|s)) = -\sum_{\tilde{a}} \pi(\tilde{a}|s) \log \pi(\tilde{a}|s)$ in the MDP:

$$\max_\pi J_P(\pi) = \mathbb{E}_{\mu,P,\pi}\left[\sum_{t=0}^\infty \gamma^t \Big(R(s_t, a_t) + \mathcal{H}(\pi(\cdot|s_t))\Big)\right] \tag{1}$$

The optimal policy satisfies the following conditions [26]:

$$\begin{aligned}
\pi(a|s) &\propto \exp\left(Q(s,a)\right) \\
Q(s,a) &= R(s,a) + \gamma \mathbb{E}_{P(s'|s,a)}\left[V(s')\right] \\
V(s) &= \log \sum_{a'} \exp\left(Q(s,a')\right)
\end{aligned} \tag{2}$$

### 2.2 Inverse Reinforcement Learning

The majority of contemporary IRL approaches have converged on the Maximum Causal Entropy (MCE) IRL framework, which aims to find a reward function $R_\theta(s,a)$ with parameters $\theta$ such that the entropy-regularized learner policy $\hat{\pi}$ has matching state-action feature with the unknown expert policy $\pi$ [27].

A related formulation casts IRL as maximum *discounted* likelihood (ML) estimation [28, 29, 30], subject to the constraint that the policy is entropy-regularized. Given a dataset of $N$ expert trajectories each of length $T$: $\mathcal{D} = \{\tau_i\}_{i=1}^N, \tau = (s_{1:T}, a_{1:T})$ sampled from the expert policy in environment $P$ with occupancy measure $\rho_\mathcal{D} := \rho_P^\pi$, ML-IRL aims to solve the following optimization problem:

$$\max_\theta \mathbb{E}_{(s_t,a_t)\sim\mathcal{D}}\left[\sum_{t=0}^T \gamma^t \log \hat{\pi}_\theta(a_t|s_t)\right], \quad \text{s.t. } \hat{\pi}_\theta(a|s) = \arg\max_{\hat{\pi}\in\Pi} \mathbb{E}_{\rho_P^{\hat{\pi}}}\left[R_\theta(s,a) + \mathcal{H}(\hat{\pi}(\cdot|s)\right] \tag{3}$$

where the policy is implicitly parameterized by the reward parameters $\theta$.

It can be shown that for sufficiently large $T \to \infty$, MCE-IRL and ML-IRL are equivalent under linear reward parameterization [28, 29], however (3) permits non-linear reward parameterization

through the following surrogate optimization problem:

$$\max_{\theta} \mathbb{E}_{\rho_{\mathcal{D}}} \left[ R_{\theta}(s,a) \right] - \mathbb{E}_{\rho_{P}^{\hat{\pi}}} \left[ R_{\theta}(s,a) \right], \quad \text{s.t. } \hat{\pi}_{\theta}(a|s) = \arg\max_{\hat{\pi} \in \Pi} \mathbb{E}_{\rho_{P}^{\hat{\pi}}} \left[ R_{\theta}(s,a) + \mathcal{H}(\hat{\pi}(\cdot|s)) \right] \quad (4)$$

(4) can be efficiently solved via alternating training of the learner policy and the reward function, similar to Generative Adversarial Network (GAN)-based algorithms [31, 32, 33, 34, 35, 36]. However, these methods all require access to the ground truth environment dynamics or a high quality simulator in order to compute or sample from the learner occupancy measure $\rho_{P}^{\hat{\pi}}$.

### 2.3 Offline Model-Based IRL & RL

Existing offline model-based IRL algorithms such as [10, 11] adapt (4) using a two-stage process. First, an estimate $\hat{P}$ of the environment dynamics is obtained from the offline dataset, e.g., using maximum likelihood estimation. Then, $\hat{P}$ is fixed and used in place of $P$ to compute $\rho_{\hat{P}}^{\hat{\pi}}$ while optimizing (4). However, this simple replacement incurs a gap between (4) and (3) which scales with the dynamics model error and the estimated value [13]. This puts a high demand on the accuracy of the estimated dynamics.

A related challenge is to prevent the policy from exploiting inaccuracies in the estimated dynamics, which can lead to erroneously high value estimates. This has been extensively studied in both online and offline model-based RL literature [37, 38, 39, 40]. The majority of recent offline model-based RL methods combat model-exploitation via a notion of "pessimism", which penalizes the learner policy from visiting states where the model is likely to be incorrect [37]. These pessimistic penalties are often designed based on quantifying uncertainty about transition dynamics through the estimated model [41, 42]. Drawing on these advances, recent offline IRL methods also incorporate pessimistic penalties into their RL subroutine [13, 12, 14]. However, designing pessimistic penalties involves nontrivial decisions to ensure that they can accurately capture out-of-distribution samples [43].

An alternative approach to avoid model-exploitation is to perform policy training against the worst-case dynamics in out-of-distribution states [44], similar to robust MDP [45, 46]. Rigter et al. [47] implemented this idea in the RAMBO algorithm and showed that it is competitive with pessimistic penalty-based approaches while requiring no penalty design and significantly less tuning. We will show that robust MDP corresponds to a sub-problem of IRL under the Bayesian formulation.

We list additional related work in Appendix A.

## 3 A Bayesian Approach to Model-based IRL

We consider model-based IRL under a Bayesian framework (BM-IRL), where the observed expert decisions are the results of the unknown reward function $R_{\theta_1}(s,a)$ and their *internal* model of the environment dynamics $\hat{P}_{\theta_2}(s'|s,a)$.

We denote the concatenated parameters with $\theta = \{\theta_1, \theta_2\}$ and condition the policy on $\theta$ as $\hat{\pi}(a|s;\theta)$ to emphasize that the expert configuration is determined by both the reward and dynamics parameters.

Upon observing a finite set of expert demonstrations $\mathcal{D}$, BM-IRL aims to compute the posterior distribution $\mathbb{P}(\theta|\mathcal{D})$ given a choice of a prior distribution $\mathbb{P}(\theta)$:

$$\mathbb{P}(\theta|\mathcal{D}) \propto \mathbb{P}(\mathcal{D}|\theta)\mathbb{P}(\theta) = \prod_{i=1}^{N} \prod_{t=1}^{T} \hat{\pi}(a_{i,t}|s_{i,t};\theta)\mathbb{P}(\theta) \quad (5)$$

where we have omitted $\prod_{i=1}^{N} \prod_{t=1}^{T} P(s_{i,t+1}|s_{i,t}, a_{i,t})$ from the likelihood because it does not depend on $\theta$.

We consider a class of prior distributions of the form:

$$\mathbb{P}(\theta) \propto \exp\left( \lambda \sum_{i=1}^{N} \sum_{t=1}^{T} \log \hat{P}_{\theta_2}(s_{i,t+1}|s_{i,t}, a_{i,t}) \right) \quad (6)$$

where the prior precision hyperparameter $\lambda$ represents how accurate we believe the expert's model of the environment is.

Let $\mathcal{L}(\theta) := \frac{1}{NT} \log \mathbb{P}(\theta|\mathcal{D})$. It can be easily verified that

$$\mathcal{L}(\theta) = \mathbb{E}_{(s,a,s')\sim\mathcal{D}} \left[ \log \hat{\pi}(a|s;\theta) + \lambda \log \hat{P}_{\theta_2}(s'|s,a) \right]$$

In this paper, we consider finding a Maximum A Posteriori (MAP) estimate of the BM-IRL model by solving the following bi-level optimization problem:

$$\max_{\theta} \mathcal{L}(\theta), \quad \text{s.t. } \hat{\pi}(a|s;\theta) = \arg\max_{\hat{\pi}\in\Pi} \mathbb{E}_{\rho_{\hat{P}}^{\hat{\pi}}} \left[ R_\theta(s,a) + \mathcal{H}(\hat{\pi}(\cdot|s)) \right] \tag{7}$$

Note that this formulation differs from (3) and the two-stage approach (see Figure 1) because it includes the log likelihood of the *internal* dynamics in the objective (weighted by $\lambda$).

It should be noted that obtaining the full posterior distribution (or an approximation) is feasible using popular approximate inference methods (e.g., stochastic variational inference or Langevin dynamics [48, 49]) and does not significantly alter the proposed estimation principles and algorithms.

## 3.1 Naive Solution

We start by presenting a naive solution to (7) which can be seen as an extension of the tabular simultaneous reward-dynamics estimation algorithms [18, 19] to the high-dimensional setting.

Solving (7) requires: 1) computing the optimal policy with respect to $\theta$, and 2) computing the gradient $\nabla_\theta \log \hat{\pi}(a|s;\theta)$ which requires inverting the policy optimization process itself. Both operations can be done exactly in the tabular setting as in prior works but are intractable in high-dimensional settings. We propose to overcome the intractability using sample-based approximation.

In this section, we focus on approximating the gradient of the policy $\nabla_\theta \log \hat{\pi}(a|s;\theta)$ and constructing a surrogate objective similar to (4) to perform simultaneous estimation. We can show that $\nabla_\theta \log \hat{\pi}(a|s;\theta)$ has the following form (see Appendix B for all proofs and derivations):

$$\nabla_\theta \log \hat{\pi}(a|s;\theta) = \nabla_\theta Q_\theta(s,a) - \mathbb{E}_{\tilde{a}\sim\hat{\pi}(\cdot|s;\theta)}[\nabla_\theta Q_\theta(s,\tilde{a})] \tag{8}$$

where $\nabla_\theta Q_\theta(s,a) = [\nabla_{\theta_1} Q_\theta(s,a), \nabla_{\theta_2} Q_\theta(s,a)]$ is the concatenation of reward and dynamics gradients defined as:

$$\nabla_{\theta_1} Q_\theta(s,a) = \mathbb{E}_{\rho_{\hat{P}}^{\hat{\pi}}(\tilde{s},\tilde{a}|s,a)} \left[ \nabla_{\theta_1} R_{\theta_1}(\tilde{s},\tilde{a}) \right] \tag{9}$$

$$\nabla_{\theta_2} Q_\theta(s,a) = \mathbb{E}_{\rho_{\hat{P}}^{\hat{\pi}}(\tilde{s},\tilde{a}|s,a)} \left[ \gamma \sum_{s'} V_\theta(s') \nabla_{\theta_2} \hat{P}_{\theta_2}(s'|\tilde{s},\tilde{a}) \right] \tag{10}$$

Given (9) and (10) are tractable to compute using sample-based approximation of expectations, we construct the following surrogate objective $\tilde{\mathcal{L}}(\theta)$ with the same gradient as the original MAP estimation problem (7):

$$\tilde{\mathcal{L}}(\theta) = \mathbb{E}_{(s,a)\sim\mathcal{D}}[\mathcal{E}_\theta(s,a)] - \mathbb{E}_{s\sim\mathcal{D},a\sim\hat{\pi}}[\mathcal{E}_\theta(s,a)] + \lambda\mathbb{E}_{(s,a,s')\sim\mathcal{D}}[\log \hat{P}_{\theta_2}(s'|s,a)] \tag{11}$$

where

$$\mathcal{E}_\theta(s,a) = \mathbb{E}_{\rho_{\hat{P}}^{\hat{\pi}}(\tilde{s},\tilde{a}|s,a)} \left[ R_{\theta_1}(\tilde{s},\tilde{a}) + \gamma EV_\theta(\tilde{s},\tilde{a}) \right], \quad EV_\theta(s,a) = \sum_{s'} \hat{P}_{\theta_2}(s'|s,a) V_\theta(s') \tag{12}$$

Optimizing (11) (subject to the same policy constraint) is now the same as optimizing (7) but tractable.

An interesting consequence of maximizing the first two terms of (11) alone (excluding the prior) is that we both increase the reward and modify the internal dynamics to generate states with higher expected value ($EV$) upon taking expert actions then following the learner policy $\hat{\pi}$, and we do the opposite when taking learner actions. Intuitively, reward and dynamics play complementary roles in determining the value of actions and thus should be regularized [50, 16, 51]. Otherwise, one cannot disentangle the effect of truly high reward and falsely optimistic dynamics. Our prior (6) alleviates this unidentifiability to some extent.

## 3.2 Robust BM-IRL

We now present our main observation that robustness emerges from the BM-IRL formulation under the dynamics accuracy prior (6).

We start by analyzing a discounted, full-trajectory version of the BM-IRL likelihood (7). Note that discounting does not change the optimal solution to (7) under expressive reward and dynamics model class; nor does it require infinite data because we can truncate the summation at $T = \text{int}\left(\frac{1}{1-\gamma}\right)$ and obtain nearly the same estimator as with infinite sequence length. Using a result from [13], we decompose the discounted likelihood as follows (see Appendix B.2):

$$
\mathbb{E}_{P(\tau)}\left[\sum_{t=0}^{\infty} \gamma^t \log \hat{\pi}(a_t|s_t; \theta)\right] = \mathbb{E}_{P(\tau)}\left[\sum_{t=0}^{\infty} \gamma^t \left(Q_\theta(s_t, a_t) - V_\theta(s_t)\right)\right]
$$

$$
= \underbrace{\mathbb{E}_{\rho_{\mathcal{D}}}\left[R_{\theta_1}(s_t, a_t)\right] - \mathbb{E}_{\mu}\left[V_\theta(s_0)\right]}_{\ell(\theta)} + \underbrace{\gamma \mathbb{E}_{\rho_{\mathcal{D}}}\left[\mathbb{E}_{s' \sim \hat{P}_{\theta_2}(\cdot|s_t, a_t)} V_\theta(s') - \mathbb{E}_{s'' \sim P(\cdot|s_t, a_t)} V_\theta(s'')\right]}_{\textbf{T1}}
$$

$$\tag{13}$$

where **T1** corresponds to the value difference under the real and estimated dynamics. We can show that **T1** is negligible if the estimated dynamics is accurate, e.g. $\mathbb{E}_{(s,a) \sim P(\tau)} D_{KL}(P(\cdot|s,a) || \hat{P}(\cdot|s,a)) \leq \epsilon$ for sufficiently small $\epsilon$, under the *expert* data distribution, which can be achieved by setting a large $\lambda$. Then, **T1** can be dropped and the discounted likelihood reduces to $\ell(\theta)$.

$\ell(\theta)$ highlights the reason why the proposed BM-IRL approach can be robust to limited dataset. It poses the offline IRL problem as maximizing the cumulative reward of expert trajectories in the real environment, and minimizing the cumulative reward generated by the learner in the estimated dynamics with respect to *both* reward and dynamics. In other words, it aims to find performance-matching reward and policy under the *worst-case* dynamics, which is trained adversarially outside the offline data distribution. This connects BM-IRL with the robust MDP approach to offline model-based RL [44, 47]. We refer to this version of BM-IRL as robust model-based IRL (RM-IRL).

## 3.3 Proposed Algorithms

We now propose two scalable algorithms to find the MAP solution to (7) via the surrogate objective (11). The first algorithm (**BM-IRL**; 1) applies the naive solution, while the second algorithm (**RM-IRL**; 2) exploits the observation in section 3.2 to derive a more efficient algorithm for high $\lambda$.

The estimation problem (7) has an inherently nested structure where, for each update of parameters $\theta$ (the outer problem), we have to solve for the optimal policy $\hat{\pi}(a|s; \theta)$ (the inner problem). Following recent ML-IRL approaches [29, 13], we perform the nested optimization using *two-timescale* stochastic approximation [52, 53], where the inner problem is solved via stochastic gradient updates on a faster time scale than the outer problem. For both algorithms, we solve the inner problem using Model-Based Policy Optimization (MBPO; [39]) which uses Soft Actor-Critic (SAC; [26]) in a dynamics model ensemble.

**BM-IRL**: For the BM-IRL outer problem, we estimate the expectations in (11) and (12) via sampling and perform coordinate-ascent optimization. Specifically, for each update step, we first sample a mini-batch of state-action pairs $(s, a) \sim \mathcal{D}$ and a mini-batch of (fake) actions $a_{\text{fake}} \sim \hat{\pi}(\cdot|s; \theta)$ and simulate both $(s, a)$ and $(s, a_{\text{fake}})$ forward in the estimated dynamics $\hat{P}$ to get the real and fake trajectories $\tau_{\text{real}}, \tau_{\text{fake}}$. We then optimize the reward function first by taking a single gradient step to optimize the following objective function:

$$
\max_{\theta_1} \quad \mathbb{E}_{(s,a) \sim \mathcal{D}, \rho_{\hat{P}}^{\hat{\pi}}(\tilde{s}, \tilde{a}|s, a)}\left[R_{\theta_1}(\tilde{s}, \tilde{a})\right] - \mathbb{E}_{s \sim \mathcal{D}, a_{\text{fake}} \sim \hat{\pi}(\cdot|s; \theta), \rho_{\hat{P}}^{\hat{\pi}}(\tilde{s}, \tilde{a}|s, a_{\text{fake}})}\left[R_{\theta_1}(\tilde{s}, \tilde{a})\right] \tag{14}
$$

Lastly, we optimize the dynamics model by taking a few gradient steps (a hyperparameter) to optimize the following objective function using on-policy rollouts branched from mini-batches of

expert state-actions as in RAMBO [47]:

$$\max_{\theta_2} \quad \lambda_1 \mathbb{E}_{(s,a)\sim\mathcal{D},\rho_{\hat{P}}^{\hat{\pi}}(\tilde{s},\tilde{a}|s,a)} \left[ EV_{\theta_2}(\tilde{s},\tilde{a}) \right] - \lambda_1 \mathbb{E}_{s\sim\mathcal{D},a_{\text{fake}}\sim\hat{\pi}(\cdot|s,;\theta),\rho_{\hat{P}}^{\hat{\pi}}(\tilde{s},\tilde{a}|s,a_{\text{fake}})} \left[ EV_{\theta_2}(\tilde{s},\tilde{a}) \right]$$
$$+ \lambda_2 \mathbb{E}_{(s,a,s')\sim\mathcal{D}} \left[ \log \hat{P}_{\theta_2}(s'|s,a) \right] \tag{15}$$

where we have introduced weighting coefficients $\lambda_1$ and $\lambda_2$ to facilitate tuning the prior precision $\lambda$ and dynamics model learning rate. We estimate the dynamics gradient using the REINFORCE method with baseline.

**RM-IRL**: We adapt the BM-IRL algorithm slightly for the RM-IRL outer problem, where we only simulate a single trajectory for each state in the mini-batch and update the reward using the following objective:

$$\max_{\theta_1} \mathbb{E}_{\rho_{\mathcal{D}}} \left[ R_{\theta_1}(s,a) \right] - \mathbb{E}_{\rho_{\hat{P}}^{\hat{\pi}}} \left[ R_{\theta_1}(s,a) \right] \tag{16}$$

For the dynamics update, we set $\lambda_2 \gg \lambda_1$ and drop the first term in (15):

$$\max_{\theta_2} \quad - \lambda_1 \mathbb{E}_{s\sim\mathcal{D},a_{\text{fake}}\sim\hat{\pi}(\cdot|s;\theta),\rho_{\hat{P}}^{\hat{\pi}}(\tilde{s},\tilde{a}|s,a_{\text{fake}})} \left[ EV_{\theta_2}(\tilde{s},\tilde{a}) \right] + \lambda_2 \mathbb{E}_{(s,a,s')\sim\mathcal{D}} \left[ \log \hat{P}_{\theta_2}(s'|s,a) \right] \tag{17}$$

We provide additional details and pseudo code for the proposed algorithms in Appendix C.

### 3.4 Performance Guarantees

In this section, we study how policy and dynamics estimation error affect learner performance in the real environment. Using lemma 4.1 from [54] which decomposes the real environment performance gap between the expert and the learner into their policy and model advantages in the estimated dynamics, we arrive at the follow performance bound (see Appendix B.3):

**Theorem 3.1.** *Let* $\epsilon_{\hat{\pi}} = -\mathbb{E}_{(s,a)\sim d_P^\pi}[\log \hat{\pi}_{\hat{P}}(a|s)]$ *be the policy estimation error and* $\epsilon_{\hat{P}} = \mathbb{E}_{(s,a)\sim d_P^\pi} D_{KL}[P(\cdot|s,a)||\hat{P}(\cdot|s,a)]$ *be the dynamics estimation error. Let* $R_{max} = \max_{s,a}|R_\theta(s,a)| + \log|\mathcal{A}|$. *Assuming bounded expert-learner marginal state-action density ratio* $\left\| \frac{d_P^{\hat{\pi}}(s,a)}{d_P^\pi(s,a)} \right\|_\infty \leq C$, *we have the following (absolute) performance bound for the IRL agent:*

$$|J_P(\hat{\pi}) - J_P(\pi)| \leq \frac{1}{1-\gamma}\epsilon_{\hat{\pi}} + \frac{\gamma(C+1)R_{max}}{(1-\gamma)^2}\sqrt{2\epsilon_{\hat{P}}} \tag{18}$$

This bound shows that the performance gap between the IRL agent and the expert is linear with respect to the discount factor in the policy estimation error and quadratic in the dynamics estimation error. Thus, ensuring small dynamics estimation error is essential and prior simultaneous estimation approaches such as [18] which do not explicitly encourage dynamics accuracy are not expected to perform well in general.

## 4 Experiments

We aim to answer the following questions with our experiments: 1) How does the dynamics accuracy prior affect agent behavior? 2) How well does BM-IRL and RM-IRL perform compared to SOTA offline IRL algorithms? We investigate Q1 using a Gridworld environment. We investigate Q2 using the standard D4RL dataset on MuJoCo continuous control benchmarks.

### 4.1 Gridworld Example

To understand the behavior of the BM-IRL algorithm, we use a 5x5 deterministic gridworld environment where the expert travels from the lower left to the upper right corner using a policy with a discount factor of $\gamma = 0.7$. We represent the reward function as the log probability of the target state: $\log \tilde{P}(s)$, where the upper right corner has a target probability of 1.

We parameterize both reward and dynamics using softmax of logits. Using 100 expert trajectories of length 50, we trained 3 BM-IRL agents with $\lambda \in \{0.001, 0.5, 10\}$. As a comparison, we also trained a two-stage IRL agent whose dynamics model was fixed after an initial maximum likelihood pretraining step and its reward was estimated using the same gradient update rule as BM-IRL in (14).

The ground truth and estimated target state probabilities are shown in the first row of Figure 2. All agents correctly estimated that the upper right corner has the highest reward, although not with the same precision as the ground truth sparse reward. BM-IRL agents with $\lambda = 0.5$ and $\lambda = 10$ were able to assign high reward only to states close to the true goal state, where as the BM-IRL agent with $\lambda = 0.001$ and the two-stage IRL agent assigned high rewards to state much further away from the true goal state.

Figure 2: Gridworld experiment results. Ground truth and estimated target state distributions (softmax of reward; **Row 1**) and sample path of estimated policy in estimated dynamics (**Row 2**) for two-stage and BM-IRL agents with $\lambda = [0.001, 0.5, 10]$. BM-IRL agents with higher $\lambda$ obtain more accurate reward estimates and commit fewer illegal transitions.

We visualize the estimated dynamics models by sampling 100 imagined rollouts using the estimated policies in the second row of Figure 2. This figure shows that the BM-IRL($\lambda = 0.001$) agent, which corresponds roughly to Herman et al.'s simultaneous estimation algorithm where no (or a very weak) prior is used [18], and the two-stage IRL agent would take significantly more illegal transitions (i.e., diagonal transitions) than BM-IRL agent with higher $\lambda$. Comparing among BM-IRL agents, we see that increasing $\lambda$ decreases the number of illegal transitions. In contrast to the two-stage IRL agent whose illegal transitions are rather random, the illegal transitions generated by BM-IRL agents with lower $\lambda$ have a strong tendency to point towards the goal state. This corroborates with our analysis that BM-IRL learns optimistic dynamics under the expert distribution.

## 4.2 MuJoCo Benchmarks

We study the performance of the proposed algorithms in the MuJoCo continuous control benchmark [55] provided by the D4RL dataset [56]. Following prior IRL evaluation protocols, our IRL agents maintain two datasets: 1) a *transition dataset* is used to trained the dynamics model and the actor-critic networks and 2) an *expert dataset* is used to train the reward function. The transition dataset is selected from one of the following: medium, medium-replay, and medium-expert, where medium-expert has the highest quality with the most complete coverage of expert trajectories. We fill the expert dataset with 10 randomly sampled D4RL expert trajectories. For each evaluation environment (HalfCheetah, Hopper, Walker2D) and transition dataset, we train our algorithms offline for a fixed number of epochs and repeat this process for 5 random seeds. After the final epoch, we evaluate the agent for 10 episodes in the corresponding environment. For both BM-IRL and RM-IRL, we set $\lambda_1 = 0.01, \lambda_2 = 1$ to encourage an accurate dynamics model under the offline data distribution. We provide additional implementation and experimental details in Appendix D.

For the baseline algorithms, we choose Behavior Cloning (BC) and ValueDICE [57], both of which are model-free offline imitation learning algorithms and they do not use the transition dataset. For

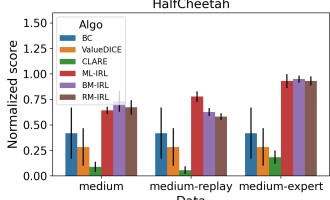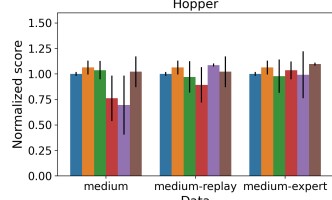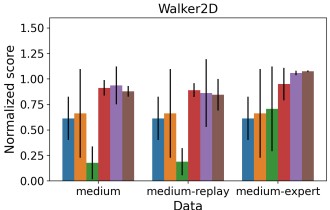

Figure 3: MuJoCo benchmark performance using 10 expert trajectories from the D4RL dataset. Bar heights and error bars represent the means and standard deviations of normalized scores, respectively, over 5 random seeds. Baseline algorithm performances are taken from [13].

SOTA offline model-based IRL algorithms, we compare with the following: 1) ML-IRL uses a similar two-timescale algorithm but operates in a two-stage fashion with a pessimistic penalty which adapts to the learner policy's state-action visitation [13], 2) CLARE performs marginal density matching with a mixture of expert and transition datasets weighted by the in-distributionness of non-expert state-action pairs [12]. We expect our algorithms to outperform CLARE whose marginal density matching method requires a larger expert dataset to work well as shown in [13, 12]. We also expect to perform similarly to ML-IRL, but better when adversarial model training is more appropriate than ensemble-based pessimistic penalty for a given dataset. Lastly, we expect BM-IRL to perform better than RM-IRL since BM-IRL solves the simultaneous estimation problem exactly.

Figure 3 shows the results of our algorithms and the comparisons reported in [13]. In the Hopper environment, all algorithms performed similarly on all datasets, except for BM-IRL and ML-IRL having lower performance with much larger variance on the medium dataset. In the HalfCheetah and Walker2D environments, our algorithms and ML-IRL performed significantly better. BM-IRL and RM-IRL outperformed other algorithms in 6/9 settings, and in 2/9 settings, the differences from the best algorithm were very small. The only case where the best algorithm, ML-IRL, significantly outperformed our algorithms was HalfCheetach medium-replay. This is mostly likely because in this environment, the medium-replay dataset has much less coverage on expert trajectories so that adversarial model training might have been too conservative.

As we expected, BM-IRL outperformed RM-IRL in 7/9 settings. The only case where its mean performance was significantly lower was Hopper medium where BM-IRL performance had large variance. However, as we explain in the section 5, this is because BM-IRL has higher training instability where its peak performance was in fact on par with RM-IRL.

# 5   Limitations

A limitation of the proposed algorithms is that BM-IRL can have less stable training dynamics than RM-IRL where its evaluation performance may alternate between periods of near optimal performance and periods of medium performance (thus the larger variance in Figure 3). While stability is a known issue for training energy-based models using contrastive divergence objectives (i.e., objective (11)) [58], we believe the current issue is related to BM-IRL's two-sample path method having weaker and noisier learning signal. Another source of instability is likely introduced by simultaneously training the dynamics model, which may be improved in future work by adding Lipschitz regularizations [59].

# 6   Conclusion

We showed that inverse reinforcement learning under a Bayesian simultaneous estimation framework gives rise to robust policies. This yielded a set of novel offline model-based IRL algorithms achieving SOTA performance in the MuJoCo continuous control benchmarks without ad hoc pessimistic penalty design. An interesting future direction is to identify appropriate priors to robustly infer reward and internal dynamics from sub-optimal and biased human demonstrators.

**Acknowledgments**

A. Garcia would like to acknowledge partial support from the Army Research Office grant W911NF-22-1-0213. M. Hong and S. Zeng are supported by NSF grant CIF-1910385. The authors would also like to acknowledge Marc Rigter for answering questions about the RAMBO algorithm.

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

# Supplemental Materials for: A Bayesian Approach to Robust Inverse Reinforcement Learning

## A    Additional Related Work

**Bayesian IRL**: Ramachandran and Amir [60] first proposed a Bayesian formulation of IRL to solve the reward ambiguity problem. A MAP inference approach was proposed in [61] and a variational inference approach was proposed in [62]. Their formulations considers non-entropy-regularized policies and the dynamics model is fixed during reward inference. In contrast, simultaneous estimation of reward and dynamics can potentially infer the demonstrator's biased beliefs about the environment, which is desirable for psychology and human-robot interaction studies [15, 19, 16]. Despite the attractiveness, simultaneous estimation is challenging because of the need to invert the agent's planning process, especially in continuous domains. Reddy et al. [16] avoids this by representing agent discrete action policies using neural network-parameterized $Q$ functions and regularizing the Bellman error to be small over the entire state-action space. This method however cannot be straightforwardly adapted to the continuous action case. Kwon et al. [63] avoids this by first training a task-conditioned policy on a distribution of environments with known parameters using meta reinforcement learning and then use the meta-trained policy to guide parameter inference. This precludes the method from being used in general settings with unknown task distributions. To our knowledge, our proposed algorithms are the first to address simultaneous estimation in general environments.

**Decision-aware model learning**: Decision-aware model learning aims to solve the objective mismatch problem in model-based RL [64]. Many proposed methods in this class use value-targeted regression similar to our model loss in (15) [65, 66]. Our analysis and that of Vemula et al. [54] suggest that value-targeted model objectives may be related to robust objectives. Furthermore, since the set of value-equivalent models only shrink for increasingly larger set of policy and values [65], using value-aware model objective alone may not be optimal and additional prediction-based regularizations may be needed.

## B    Missing Proofs

### B.1    Proofs For Section 3.1

**Derivation of BM-IRL Gradients (section 3.1).** Recall the definition of the optimal entropy-regularized policy and value functions:

$$
\begin{aligned}
\hat{\pi}(a|s; \theta) &= \frac{\exp(Q_\theta(s, a))}{\sum_{\tilde{a}} \exp(Q_\theta(s, \tilde{a}))} \\
Q_\theta(s, a) &= R_{\theta_1}(s, a) + \gamma \mathbb{E}_{s' \sim \hat{P}_{\theta_2}(\cdot|s,a)}[V_\theta(s')] \\
V_\theta(s) &= \log \sum_{\tilde{a}} \exp(Q_\theta(s, \tilde{a}))
\end{aligned}
\tag{19}
$$

The gradient of the policy log likelihood in terms of the $Q$ function gradient is obtained as follows:

$$
\begin{aligned}
\nabla_\theta \log \hat{\pi}(a|s; \theta) &= \nabla_\theta Q_\theta(s, a) - \nabla_\theta V_\theta(s) \\
&= \nabla_\theta Q_\theta(s, a) - \frac{1}{Z_\theta} \nabla_\theta \sum_{\tilde{a}} \exp(Q_\theta(s, \tilde{a})) \\
&= \nabla_\theta Q_\theta(s, a) - \frac{1}{Z_\theta} \sum_{\tilde{a}} \exp(\nabla_\theta Q_\theta(s, \tilde{a})) \\
&= \nabla_\theta Q_\theta(s, a) - \mathbb{E}_{\tilde{a} \sim \hat{\pi}(\cdot|s;\theta)}[\nabla_\theta Q_\theta(s, \tilde{a})]
\end{aligned}
\tag{20}
$$

where $Z_\theta = \sum_{a'} \exp(Q_\theta(s, a'))$ is the normalizer.

Recall $\rho_{\hat{P}}^{\hat{\pi}}(\tilde{s}, \tilde{a}|s,a)$ is the discounted state-action occupancy measure starting from pair $(s,a)$. We define for any function $f(s,a)$:

$$\mathbb{E}_{\rho_{\hat{P}}^{\hat{\pi}}(\tilde{s},\tilde{a}|s,a)}[f(s,a)] = \mathbb{E}_{\tau \sim (\hat{P},\hat{\pi})}\left[\sum_{t=0}^{\infty} \gamma^t f(s,a)\Big| s_0 = s, a_0 = a\right] \tag{21}$$

We now derive $Q$ function gradients with respect to the reward parameters $\theta_1$ and dynamics parameters $\theta_2$, respectively.

$$
\begin{aligned}
\nabla_{\theta_1} Q_\theta(s,a) &= \nabla_{\theta_1} R_{\theta_1}(s,a) + \gamma \mathbb{E}_{s' \sim \hat{P}_{\theta_2}(\cdot|s,a)}[\nabla_{\theta_1} V_\theta(s')] \\
&= \nabla_{\theta_1} R_{\theta_1}(s,a) + \gamma \mathbb{E}_{s' \sim \hat{P}_{\theta_2}(\cdot|s,a), a' \sim \hat{\pi}(\cdot|s';\theta)}[\nabla_{\theta_1} Q_\theta(s',a')] \\
&= \nabla_{\theta_1} R_{\theta_1}(s,a) + \gamma \mathbb{E}_{s' \sim \hat{P}_{\theta_2}(\cdot|s,a), a' \sim \hat{\pi}(\cdot|s';\theta)}\Bigg[ \\
&\quad \nabla_{\theta_1} R_{\theta_1}(s',a') + \gamma \mathbb{E}_{s'' \sim \hat{P}_{\theta_2}(\cdot|s',a'), a'' \sim \hat{\pi}(\cdot|s'';\theta)}[\nabla_{\theta_1} Q_\theta(s'',a'')]\Bigg] \\
&= \nabla_{\theta_1} R_{\theta_1}(s,a) + \mathbb{E}_{\tau \sim (\hat{P},\hat{\pi})}\left[\sum_{h=1}^{\infty} \gamma^h \nabla_{\theta_1} R_{\theta_1}(s_h, a_h)\Big| s_0 = s, a_0 = a\right] \\
&= \mathbb{E}_{\rho_{\hat{P}}^{\hat{\pi}}(\tilde{s},\tilde{a}|s,a)}[\nabla_{\theta_1} R_{\theta_1}(\tilde{s},\tilde{a})]
\end{aligned}
\tag{22}
$$

In line two we used the result that $\nabla_\phi V_\theta(s)$ for both $\phi = \theta_1$ and $\phi = \theta_2$ corresponds to the second term in (20) .

$$
\begin{aligned}
\nabla_{\theta_2} Q_\theta(s,a) &= \nabla_{\theta_2} R_{\theta_1}(s,a) + \nabla_{\theta_2} \gamma \mathbb{E}_{s' \sim \hat{P}_{\theta_2}(\cdot|s,a)}[V_\theta(s')] \\
&= \gamma \sum_{\tilde{s}} V_\theta(\tilde{s}) \nabla_{\theta_2} \hat{P}_{\theta_2}(\tilde{s}|s,a) + \gamma \mathbb{E}_{s' \sim \hat{P}_{\theta_2}(\cdot|s,a), a' \sim \hat{\pi}(\cdot|s';\theta)}[\nabla_{\theta_2} Q_\theta(s',a')] \\
&= \gamma \sum_{\tilde{s}} V_\theta(\tilde{s}) \nabla_{\theta_2} \hat{P}_{\theta_2}(\tilde{s}|s,a) + \gamma \mathbb{E}_{s' \sim \hat{P}_{\theta_2}(\cdot|s,a), a' \sim \hat{\pi}(\cdot|s';\theta)}\Bigg[ \\
&\quad \gamma \sum_{\tilde{s}} V_\theta(\tilde{s}) \nabla_{\theta_2} \hat{P}_{\theta_2}(\tilde{s}|s',a') + \gamma \mathbb{E}_{s'' \sim \hat{P}_{\theta_2}(\cdot|s',a'), a'' \sim \hat{\pi}(\cdot|s'';\theta)}[\nabla_{\theta_2} Q_\theta(s'',a'')]\Bigg] \\
&= \gamma \sum_{\tilde{s}} V_\theta(\tilde{s}) \nabla_{\theta_2} \hat{P}_{\theta_2}(\tilde{s}|s,a) + \mathbb{E}_{\tau \sim (\hat{P},\hat{\pi})}\left[\sum_{h=1}^{\infty} \gamma^{h+1} \sum_{\tilde{s}} V_\theta(\tilde{s}) \nabla_{\theta_2} \hat{P}_{\theta_2}(\tilde{s}|s_h, a_h)\Big| s_0 = s, a_0 = a\right] \\
&= \mathbb{E}_{\rho_{\hat{P}}^{\hat{\pi}}(\tilde{s},\tilde{a}|s,a)}\left[\gamma \sum_{s'} V_\theta(s') \nabla_{\theta_2} \hat{P}_{\theta_2}(s'|\tilde{s},\tilde{a})\right]
\end{aligned}
\tag{23}
$$

We make a quick remark on the identifiability of simultaneous estimation.

**Remark B.1.** *Simultaneous reward-dynamics estimation of the form (5) without specific assumptions on the prior $P(\theta)$ is in general unidentifiable.*

*Proof.* Let $\mathbf{R} \in \mathbb{R}^{|\mathcal{S}||\mathcal{A}|}$ and $\mathbf{P} \in \mathbb{R}_+^{|\mathcal{S}||\mathcal{A}| \times |\mathcal{S}|}$, $\sum_{s'} \mathbf{P}_{ss'}^a = 1$, $\mathbf{Q} \in \mathbb{R}^{|\mathcal{S}||\mathcal{A}|}$ and $\mathbf{V} \in \mathbb{R}^{|\mathcal{S}|}$ be a set of Bellman-consistent reward, dynamics, and value functions in matrix form. Let $\mathbf{P}' \neq \mathbf{P}$ be an alternative dynamics model. We can always find an alternative reward $\mathbf{R}' = \mathbf{R} + \Delta\mathbf{R}$, where:

$$
\begin{aligned}
\Delta\mathbf{R} &= (\mathbf{Q} - \mathbf{Q}) - \gamma(\mathbf{P}'\mathbf{V} - \mathbf{P}\mathbf{V}) \\
&= -\gamma\Delta\mathbf{P}\mathbf{V}
\end{aligned}
\tag{24}
$$

without changing the value functions and optimal entropy-regularized policy. $\qquad\square$

Remark B.1 implies that existing simultaneous estimation approaches which do not use explicit or implicit regularizations, such as the SERD algorithm by [18], cannot in general accurately estimate expert reward. Paired with theorem 3.1, it shows that these algorithms cannot in general achieve good performance.

## B.2 Proofs For Section 3.2

**Derivation of discounted likelihood (13).**

$$
\mathbb{E}_{P(\tau)}\left[\sum_{t=0}^{\infty}\gamma^t \log \hat{\pi}(a_t|s_t;\theta)\right]
$$

$$
= \mathbb{E}_{P(\tau)}\left[\sum_{t=0}^{\infty}\gamma^t \left(Q_\theta(s_t,a_t) - V_\theta(s_t)\right)\right]
$$

$$
= \mathbb{E}_{P(\tau)}\left[\sum_{t=0}^{\infty}\gamma^t \left(R_{\theta_1}(s_t,a_t) + \gamma\mathbb{E}_{s'\sim\hat{P}_{\theta_2}(\cdot|s_t,a_t)}[V_\theta(s')]\right)\right] - \mathbb{E}_{P(\tau)}\left[\sum_{t=0}^{\infty}\gamma^t V_\theta(s_t)\right]
$$

$$
= \mathbb{E}_{P(\tau)}\left[\sum_{t=0}^{\infty}\gamma^t R_{\theta_1}(s_t,a_t)\right] - \mathbb{E}_\mu\left[V_\theta(s_0)\right]
$$

$$
+ \mathbb{E}_{P(\tau)}\left[\sum_{t=0}^{\infty}\gamma^{t+1}\mathbb{E}_{s'\sim\hat{P}_\theta(\cdot|s_t,a_t)}[V_\theta(s')]\right] - \mathbb{E}_{P(\tau)}\left[\sum_{t=1}^{\infty}\gamma^t V_\theta(s_t)\right]
$$

$$
= \mathbb{E}_{P(\tau)}\left[\sum_{t=0}^{\infty}\gamma^t R_{\theta_1}(s_t,a_t)\right] - \mathbb{E}_\mu\left[V_\theta(s_0)\right]
$$

$$
+ \mathbb{E}_{P(\tau)}\left[\sum_{t=0}^{\infty}\gamma^{t+1}\mathbb{E}_{s'\sim\hat{P}_\theta(\cdot|s_t,a_t)}[V_\theta(s')]\right] - \mathbb{E}_{P(\tau)}\left[\sum_{t=0}^{\infty}\gamma^{t+1}\mathbb{E}_{s'\sim P(\cdot|s_t,a_t)}[V_\theta(s')]\right]
$$

$$
= \underbrace{\mathbb{E}_{\rho_P^\pi}\left[R_{\theta_1}(s_t,a_t)\right] - \mathbb{E}_\mu\left[V_\theta(s_0)\right]}_{\ell(\theta)} + \underbrace{\gamma\mathbb{E}_{\rho_P^\pi}\left[\mathbb{E}_{s'\sim\hat{P}_\theta(\cdot|s_t,a_t)}V_\theta(s') - \mathbb{E}_{s''\sim P(\cdot|s_t,a_t)}V_\theta(s'')\right]}_{\textbf{T1}}
$$

$$(25)$$

The following lemma shows that **T1** is negligible if the estimated dynamics is accurate under the *expert* distribution, which is available from the offline dataset.

**Lemma B.2.** *Let* $\epsilon = \mathbb{E}_{(s,a)\sim P(\tau)}D_{KL}(P(\cdot|s,a)||\hat{P}(\cdot|s,a))$ *and* $R_{max} = \max_{s,a}|R_\theta(s,a)| + \log|\mathcal{A}|$, *it holds that*

$$
|\textbf{T1}| \leq \frac{\gamma R_{max}}{(1-\gamma)^2}\sqrt{2\epsilon} \tag{26}
$$

*Proof.*

$$
|\textbf{T1}| = \left|\sum_{t=0}^{\infty}\gamma^{t+1}\mathbb{E}_{(s_t,a_t)\sim P(\tau)}\left[\sum_{s'}V_\theta(s')\left(\hat{P}(s'|s_t,a_t) - P(s'|s_t,a_t)\right)\right]\right|
$$

$$
\overset{(1)}{\leq} \sum_{t=0}^{\infty}\gamma^{t+1}\mathbb{E}_{(s_t,a_t)\sim P(\tau)}\left[\sum_{s'}|V_\theta(s')|\left|\hat{P}(s'|s_t,a_t) - P(s'|s_t,a_t)\right|\right]
$$

$$
\overset{(2)}{\leq} \sum_{t=0}^{\infty}\gamma^{t+1}\|V_\theta(\cdot)\|_\infty\mathbb{E}_{(s_t,a_t)\sim P(\tau)}\left[\left\|\hat{P}(\cdot|s_t,a_t) - P(\cdot|s_t,a_t)\right\|_1\right]
$$

$$
\overset{(3)}{\leq} \sum_{t=0}^{\infty}\gamma^{t+1}\|V_\theta(\cdot)\|_\infty\sqrt{2\mathbb{E}_{(s_t,a_t)\sim P(\tau)}D_{KL}(P||\hat{P})}
$$

$$
= \frac{\gamma}{1-\gamma}\|V_\theta(\cdot)\|_\infty\sqrt{2\epsilon}
$$

where (1) follows from Jensen's inequality, (2) follows from Holder's inequality, and (3) follows from Pinsker's inequality.

Finally, given $\mathcal{H}(\pi(\cdot|s)) = -\sum_a \pi(a|s) \log \pi(a|s) \leq -\sum_a \pi(a|s) \log \frac{1}{|\mathcal{A}|} = \log|\mathcal{A}|$, we have $\|V_\theta(\cdot)\|_\infty \leq \mathbb{E}\left[\sum_{t=0}^{\infty} \gamma^t \left(\max_{s,a}|R_\theta(s,a)| + \log|\mathcal{A}|\right)\right] = \frac{R_{\max}}{1-\gamma}$.

$\square$

## B.3 Proofs For Section 3.4

We first restate a slight modification of the result from [54], which decomposes the real environment performance gap between the expert and the learner into their policy and model advantages in the estimated dynamics:

**Lemma B.3.** *(Performance difference via advantage in model; Lemma 4.1 in [54]) Let $d_P^\pi$ denote the marginal state-action distribution following policy $\pi$ in environment $P$. The following relationship holds:*

$$\mathbb{E}_{(s,a)\sim d_P^\pi}\left[\log \hat{\pi}_{\hat{P}}(a|s)\right] = \mathbb{E}_{s\sim d_P^\pi}\left[\mathbb{E}_{a\sim\pi}Q_{\hat{P}}^{\hat{\pi}}(s,a) - V_{\hat{P}}^{\hat{\pi}}(s)\right] \tag{27}$$

$$= \underbrace{(1-\gamma)\mathbb{E}_{s\sim\mu}\left[V_P^\pi(s) - V_{\hat{P}}^{\hat{\pi}}(s)\right]}_{\text{Performance difference in real environment}} \tag{28}$$

$$+ \underbrace{\gamma\mathbb{E}_{(s,a)\sim d_P^{\hat{\pi}}}\left[\mathbb{E}_{s'\sim P}V_{\hat{P}}^{\hat{\pi}}(s') - \mathbb{E}_{s''\sim\hat{P}}V_{\hat{P}}^{\hat{\pi}}(s'')\right]}_{\text{Model disadvantage under learner distribution}} \tag{29}$$

$$+ \underbrace{\gamma\mathbb{E}_{(s,a)\sim d_P^\pi}\left[\mathbb{E}_{s'\sim\hat{P}}V_{\hat{P}}^{\hat{\pi}}(s') - \mathbb{E}_{s''\sim P}V_{\hat{P}}^{\hat{\pi}}(s'')\right]}_{\text{Model advantage under expert distribution}} \tag{30}$$

The performance bound in theorem 3.1 can be obtained from lemma B.3 as follow:

**Theorem B.4.** *(Restate of theorem 3.1) Let $\epsilon_{\hat{\pi}} = -\mathbb{E}_{(s,a)\sim d_P^\pi}[\log \hat{\pi}_{\hat{P}}(a|s)]$ be the policy estimation error and $\epsilon_{\hat{P}} = \mathbb{E}_{(s,a)\sim d_P^\pi}D_{KL}[P(\cdot|s,a)||\hat{P}(\cdot|s,a)]$ be the dynamics estimation error. Let $R_{max} = \max_{s,a}|R_\theta(s,a)| + \log|\mathcal{A}|$. Assuming bounded expert-learner marginal state-action density ratio $\left\|\frac{d_P^{\hat{\pi}}(s,a)}{d_P^\pi(s,a)}\right\|_\infty \leq C$, we have the following (absolute) performance bound for the IRL agent:*

$$|J_P(\hat{\pi}) - J_P(\pi)| \leq \frac{1}{1-\gamma}\epsilon_{\hat{\pi}} + \frac{\gamma(C+1)R_{max}}{(1-\gamma)^2}\sqrt{2\epsilon_{\hat{P}}} \tag{31}$$

*Proof.*

$$|J_P(\hat{\pi}) - J_P(\pi)| = \left|\mathbb{E}_{s\sim\mu}\left[V_P^{\hat{\pi}}(s) - V_P^\pi(s)\right]\right|$$

$$\leq \frac{1}{1-\gamma}\epsilon_{\hat{\pi}}$$

$$+ \frac{\gamma}{1-\gamma}\mathbb{E}_{(s,a)\sim d_P^\pi}\left[\left|\frac{d_P^{\hat{\pi}}(s,a)}{d_P^\pi(s,a)}\left(\mathbb{E}_{s'\sim P}V_{\hat{P}}^{\hat{\pi}}(s') - \mathbb{E}_{s''\sim\hat{P}}V_{\hat{P}}^{\hat{\pi}}(s'')\right)\right|\right]$$

$$+ \frac{\gamma}{1-\gamma}\mathbb{E}_{(s,a)\sim d_P^\pi}\left[\left|\mathbb{E}_{s'\sim\hat{P}}V_{\hat{P}}^{\hat{\pi}}(s') - \mathbb{E}_{s''\sim P}V_{\hat{P}}^{\hat{\pi}}(s'')\right|\right]$$

$$\leq \frac{1}{1-\gamma}\epsilon_{\hat{\pi}} \tag{32}$$

$$+ \frac{\gamma}{1-\gamma}\left\|\frac{d_P^{\hat{\pi}}(\cdot,\cdot)}{d_P^\pi(\cdot,\cdot)}\right\|_\infty \left\|V_{\hat{P}}^{\hat{\pi}}(\cdot)\right\|_\infty \mathbb{E}_{(s,a)\sim d_P^\pi}\left[\left\|\hat{P}(\cdot|s,a) - P(\cdot|s,a)\right\|_1\right]$$

$$+ \frac{\gamma}{1-\gamma}\left\|V_{\hat{P}}^{\hat{\pi}}(\cdot)\right\|_\infty \mathbb{E}_{(s,a)\sim d_P^\pi}\left[\left\|\hat{P}(\cdot|s,a) - P(\cdot|s,a)\right\|_1\right]$$

$$= \frac{1}{1-\gamma}\epsilon_{\hat{\pi}} + \frac{\gamma(C+1)R_{\max}}{(1-\gamma)^2}\sqrt{2\epsilon_{\hat{P}}}$$

where the last line uses results from lemma B.2.

$\square$

## C   Further Algorithm Details and Pseudo Code

We estimate the dynamics gradient in (15) and (17) using the REINFORCE method with baseline:

$$\nabla_{\theta_2} EV_\theta(s,a) = \sum_{s'} V_\theta(s') \nabla_{\theta_2} \hat{P}_{\theta_2}(s'|s,a)$$

$$= \mathbb{E}_{s' \sim \hat{P}(\cdot|s,a)} \left[ (V_\theta(s') - b(s,a)) \nabla_{\theta_2} \log \hat{P}_{\theta_2}(s'|s,a) \right]$$

Following Rigter et al. [47], we set the baseline to $b(s,a) = Q_\theta(s,a) - R_{\theta_1}(s,a)$ to reduce gradient variance and further normalize $V_\theta(s') - b(s,a)$ across the mini-batch to stabilize training. In the continuous-control setting, the value function can be estimated as $V_\theta(s) = \mathbb{E}_{a \sim \hat{\pi}_\theta}[Q_\theta(s,a) - \log \hat{\pi}(a|s;\theta)]$ with a single sample. We apply this gradient for a fixed number of steps for dynamics model training, which is a hyperparameter.

Pseudo code for the proposed algorithms are listed in Algorithm 1 and Algorithm 2.

---

**Algorithm 1** Bayesian Model-based IRL (BM-IRL)

---

**Require:** Dataset $\mathcal{D} = \{\tau\}$, dynamics model $\hat{P}_{\theta_2}(s'|s,a)$, reward model $R_{\theta_1}(s,a)$, hyperparameters $\lambda_1, \lambda_2$
1: **for** $k = 1:K$ **do**
2:     Run MBPO to update learner policy $\hat{\pi}(a|s;\theta)$ and value function $Q_\theta(s,a)$ in dynamics $\hat{P}$
3:     Sample real trajectory $\tau_{\text{real}}$ starting from $(s,a) \sim \mathcal{D}$ and following $\hat{P}$ and $\hat{\pi}$
4:     Sample fake trajectory $\tau_{\text{fake}}$ starting from $s \sim \mathcal{D}$, $a_{\text{fake}} \sim \hat{\pi}(\cdot|s;\theta)$ and following $\hat{P}$ and $\hat{\pi}$
5:     Sample $(s,a,s') \sim \mathcal{D}$ for dynamics model training
6:     Evaluate (14) and take a gradient step
7:     Evaluate (15) and take a few gradient steps.
8: **end for**

---

**Algorithm 2** Robust Model-based IRL (RM-IRL)

---

**Require:** Dataset $\mathcal{D} = \{\tau\}$, dynamics model $\hat{P}_{\theta_2}(s'|s,a)$, reward model $R_{\theta_1}(s,a)$, hyperparameters $\lambda_1, \lambda_2$
1: **for** $k = 1:K$ **do**
2:     Run MBPO to update learner policy $\hat{\pi}(a|s;\theta)$ and value function $Q_\theta(s,a)$ in dynamics $\hat{P}$
3:     Sample real trajectory $\tau_{\text{real}} \sim \mathcal{D}$
4:     Sample fake trajectory $\tau_{\text{fake}}$ starting from $s \sim \mathcal{D}$ and following $\hat{P}$ and $\hat{\pi}$
5:     Sample $(s,a,s') \sim \mathcal{D}$ for dynamics model training
6:     Evaluate (16) and take a gradient step
7:     Evaluate (17) and take a few gradient steps
8: **end for**

---

## D   Implementation Details

Our implementation[2] builds on top of the official RAMBO implementation[3] [47].

### D.1   MuJoCo Benchmarks

For the MuJoCo benchmarks described in section 4.2, we follow standard practices in model-based RL.

---

[2]https://github.com/rw422scarlet/bmirl_tf
[3]https://github.com/marc-rigter/rambo

### D.1.1 Dynamics Pre-training

We use an ensemble of $K = 7$ neural networks where each network outputs the mean and covariance parameters of a Gaussian distribution over the difference between the next state and the current state $\delta = s' - s$:

$$\hat{P}_{\theta_2}^{(k)}(\delta|s,a) = \mathcal{N}(\delta|\mu_{\theta_2}^{(k)}(s,a), \Sigma_{\theta_2}^{(k)}(s,a)) \tag{33}$$

Each network is a 4-layer feedforward network with 200 hidden units and Sigmoid linear unit (SiLU) activation function. For the initial pre-training step, we maximize the likelihood of dataset transitions using a batch size of 256 and early stop when all models stop improving for more than 1 percent. We then select the 5 best models in terms of mean-squared-error on a 10 % holdout validation set. During model rollouts, we randomly pick one of the 5 best models (elites) to sample the next state.

Table 1: Shared hyperparameters across different environments

| | Hyparameter | BM-IRL | RM-IRL |
|---|---|---|---|
| SAC + MBPO | critic learning rate | 3e-4 | 3e-4 |
| | actor learning rate | 3e-4 | 3e-4 |
| | discount factor ($\gamma$) | 0.99 | 0.99 |
| | soft target update parameter ($\tau$) | 5e-3 | 5e-3 |
| | target entropy | -dim(A) | -dim(A) |
| | minimum temperature ($\alpha$) | 0.1 | 0.001 |
| | batch size | 256 | 256 |
| | real ratio | 0.5 | 0.5 |
| | model retain epochs | 5 | 5 |
| | training epochs | 500 | 300 |
| | steps per epoch | 1000 | 1000 |
| Dynamics | # model networks | 7 | 7 |
| | # elites | 5 | 5 |
| | adv. rollout batch size | 1000 | 256 |
| | adv. rollout steps | 10 | 10 |
| | adv. update steps | 50 | 50 |
| | adv. loss weighting ($\lambda_1$) | 0.01 | 0.01 |
| | supervised. loss weighting ($\lambda_2$) | 1 | 1 |
| | learning rate | 1e-4 | 1e-4 |
| | adv. update steps | 50 | 50 |
| Reward | max reward | 10 | 10 |
| | rollout batch size | 1000 | 64 |
| | rollout steps | 40 | 100 |
| | l2 penalty | 1e-3 | 1e-3 |
| | learning rate | 1e-4 | 1e-4 |
| | update steps | 1 | 1 |

Table 2: Environment-specific hyperparameters

| Environment | Hyperparameter | BM-IRL |
|---|---|---|
| HalfCheetah | model rollout batch size | 50000 |
| | model rollout steps | 5 |
| | model rollout frequency | 250 |
| Hopper | model rollout batch size | 10000 |
| | model rollout steps | 40 |
| | model rollout frequency | 250 |
| Walker2d | model rollout batch size | 10000 |
| | model rollout steps | 40 |
| | model rollout frequency | 250 |

### D.1.2 Policy Training

Our policy training process follows MBPO [39] which uses SAC with automatic temperature tuning [67]. Shared hyperparameters across different environments are listed in Table 1 and environment-specific hyperparameters are listed in Table 2. For the actor and critic, we use feedforward neural networks with 2 hidden layers of 256 units and ReLU activation. We train the actor and critic networks using a combination of real and simulated samples. We use a real ratio of 0.5, which is standard practice in model-based RL and IRL. We found that BM-IRL requires a higher minimum temperature to stablize training, which is set to $\alpha = 0.1$.

We found that different MuJoCo environments require different model rollout hyperparameters, similar to what's reported in [43]. Specifically, Hopper and Walker2d only work with significantly larger rollout steps. We decrease their rollout batch size to reduce computational overhead. HalfCheetah on the other hand works better with smaller rollout steps and larger rollout batch size. In contrast to Lu et al. [43], we did not use different rollout hyperparameters for different datasets.

### D.1.3 Reward and Dynamics Training

We use 10 random trajectories from the D4RL MuJoCo expert dataset after removing all expert trajectories that resulted in terminal states.

We use the same network architecture as the actor-critic to parameterize the reward function. We further clip the reward function to a maximum range of $\pm 10$ and apply l2 regularization on all weights with a penalty of 0.001.

As described in the main text, we update the reward function by simulating sample trajectories and taking a single gradient step. For RM-IRL, we randomly sample expert trajectory segments of length "rollout steps" and use the first step as the start of our simulated sample paths.

We update the dynamics using on-policy rollouts branched from the dataset state-actions. We use the same batch size for reward and dynamics rollouts, which is 1000 for BM-IRL and 256 for RM-IRL. Because only the first step in BM-IRL's real sample paths come from the dataset, it requires a larger batch size to iterate more data samples. We also train BM-IRL for more epochs than RM-IRL.

To compute the dynamics log likelihood in the REINFORCE gradient in (33), we treat the ensemble as a uniform mixture and compute the likelihood as:

$$\hat{P}_{\theta_2}(\delta|s,a) = \frac{1}{K} \sum_{k=1}^{K} \hat{P}_{\theta_2}^{(k)}(\delta|s,a) \tag{34}$$

We set the dynamics adversarial loss weighting to $\lambda_1 = 0.01$ for both BM-IRL and RM-IRL. We found this to work better than what's in the official RAMBO implementation, which is $\lambda_1 = 0.0768$. Note that the RAMBO author reported $\lambda_1 = 3e\text{-}4$ in their paper but forget to average their REINFORCE loss over the mini-batch of size 256 in their implementation, which is instead treated as a sum by default by TensorFlow. We empirically found that small $\lambda_1$ leads to severe model exploitation.

