# OpenReview forum: "A Bayesian Approach to Robust Inverse Reinforcement Learning"
_robot-learning.org/CoRL/2023/Conference — CoRL 2023 Poster_

### Official Review · Reviewer_agyf · 2023-07-19

**Confidence:** 4
**Originality:** Good
**Technical Quality:** Very Good
**Clarity Of Presentation:** Good
**Impact:** 4

**Recommendation:**

Weak Accept: I recommend accepting the paper, but will not argue for my recommendation if the majority of other reviewers have a different opinion.

**Review:**

The paper addresses an important challenge in IRL: how to mitigate compounding errors in the traditional two-step approaches where dynamic estimates are obtained prior to reward and policy learning. The paper is well-written and the idea of considering IRL under the Bayesian Theory of Mind framework is well-motivated and interesting.

**Quality Of The Limitations Section:**

Limitations are addressed clearly

**Questions For Rebuttal:**

Although the evaluation does a good job of showcasing the performance and advantages of the proposed framework, there are several points for which additional information or details are missing:

1. Figure 1 shows that although agents with higher confidence in the accuracy of the expert's model result in better dynamics estimates, illegal transitions are still present. What are the implications of these results for reward and policy learning? Also, does it mean that a higher lambda will help to mitigate these illegal transitions?
2. What lambda values were used for the MuJoCO benchmark and how should this value be chosen in practice?
3. From Table 1, why does ML-IRL achieve better results for the medium replay dataset in the HalfCheetah and Walker2D?
4. Table 1 shows the quality of the demonstrations can affect the performance of BTOM and RTOM. A more detailed analysis that shows the trade-offs and correlations between the choice of lambda and the coverage of expert demonstrations should be included and discussed in detail.

**Robotics Focus:**

Relevant but unlikely to deploy to hardware in near future

**Summary Of Paper:**

The paper proposes an offline inverse reinforcement learning (IRL) framework in which both the reward and transition functions are learned simultaneously from observations in an offline manner. The proposed framework is built on the Bayesian Theory of Mind perspective under which expert demonstrations implicitly convey their beliefs about the environment dynamics. Using this idea, the authors propose a class of priors that encode how accurate the learning agent believes the expert model of the environment is and later show how variations in this prior affect the learner agent's performance. Two solutions (naive and robot) to this new formulation are also proposed and evaluated in simulation.


**Summary Of Recommendation:**

I think the core idea is interesting and applicable to varied domains such as robotics and psychology. However, I would like to see a more thorough analysis and/or discussion of how BTOM and RTOM performance changes when there is a discrepancy between the learning agent's belief and the accuracy of the expert's internal dynamics model.

---

### Official Review · Reviewer_zKpN · 2023-07-19

**Confidence:** 3
**Originality:** Good
**Technical Quality:** Very Good
**Clarity Of Presentation:** Good
**Impact:** 3

**Recommendation:**

Weak Accept: I recommend accepting the paper, but will not argue for my recommendation if the majority of other reviewers have a different opinion.

**Review:**

Strengths:
* Using Bayesian Theory of Mind as a motivating framework for model-based inverse reinforcement learning is an interesting perspective. The mapping of the dynamics model to belief and reward to desire is intuitive on its face and differs from that in prior work (although I also feel that this is a bit of aweakness, see below).

* The derivations appear to be correct to my eye, and the connection between the proposed method and prior robust model-based offline RL methods is a nice observation. Intuitively this connection makes sense and provides context for the methods.

Weaknesses:
* While an interesting perspective, I also feel that there is an over-emphasis on the Bayesian Theory of Mind aspect to a point where it detracts from the paper. Although the dynamics model and reward can be mapped to the beliefs and desires of an agent, this seems to me to simply be a motivation for construction a Bayesian posterior (in Eq. 5) as opposed to a full theory of mind model. For example, this work appears to assume that the expert's model of the environment dynamics is stationary, i.e., maps to a fixed belief, as opposed to rigorous theory of mind models which often consider dynamic beliefs in partially observable (POMDP) environments.

* The relationship to prior methods is not entirely clear. Line 44 indicates that the proposed algorithms are extensions of prior works to continuous spaces, but this connection is never touched on again. The loss function defined in Eq. 7 is clearly similar to that defined in [18] (plus an entropy regularizer), so is the primary contribution the sample-based approximation scheme introduced in Sec. 3.1? Does the proposed Bayesian setup contribute to this, or could the sample-based approximations also have been directly applied to the loss functions defined in prior work?

* The experimental results are not very comprehensive, as the proposed methods are only compared to a single prior method (ML-IRL). Given that connections are drawn to robust methods (specifically RAMBO), I would have expected a comparison to at least this method if not others. In addition, the results seem somewhat ambiguous. In some scenarios BTOM performs best, while others RTOM best, and others ML-IRL. Some analysis as to why this is the case would be helpful, e.g., in what scenarios is RTOM/ML-IRL expected to perform better than BTOM.

**Quality Of The Limitations Section:**

Limitations are addressed clearly

**Questions For Rebuttal:**

Questions:
1) What does BTOM and RTOM stand for? They are not defined anywhere (presumably BTOM is Bayesian Theory of Mind which is also a bit confusing as that is an existing model).

2) I'm not sure I totally understand the bottom row of Fig. 1. What is the input and output of the dynamics model? I am confused as to why the generated transitions are stochastic (ground truth figure) when the gridworld is a 5x5 environment with discrete actions and deterministic transitions. I would have expected a perfect grid for the ground truth dynamics model.

3) Line 255 indicates that the results for ML-IRL are pulled from the paper [13], but this does not appear to be the case as the values don't match. Are these re-generated?

4) How sensitive is lambda to the algorithm performance? Does this value require tuning?

**Robotics Focus:**

Highly relevant to robotics but no hardware experiments

**Summary Of Paper:**

This paper introduces a model-based inverse reinforcement learning approach for offline datasets, in which a dynamics model and reward function are estimated simultaneously. The method is motivated by a Bayesian Theory of Mind framework and approaches the problem from a bayesian inference perspective (MAP estimate). Two variants are proposed, BTOM and RTOM, where BTOM solves the full solution and RTOM solves a reduced version under the assumption that the real and estimated dynamics are sufficiently close. Experiments in Gridworld and MuJoCo are performed in which the proposed methods are compared to a prior method, ML-IRL, and show performance improvements in some scenarios.

**Summary Of Recommendation:**

As the paper currently stands, I would lean towards a weak reject due to the lack of baselines in the experimental results and the lack of clarity in the methodology. I am open to revising my recommendation though given any additional updates in the rebuttal as I think the paper has a lot of potential.

---

### Official Review · Reviewer_oFYH · 2023-07-21

**Confidence:** 5
**Originality:** Very Good
**Technical Quality:** Good
**Clarity Of Presentation:** Fair
**Impact:** 4

**Recommendation:**

Weak Accept: I recommend accepting the paper, but will not argue for my recommendation if the majority of other reviewers have a different opinion.

**Review:**

# Strengths

## Quality

The theoretical results in the paper are valuable contributions. I really appreciate the connection to robustness that they identify. I also appreciate the discussion of performance guarantees.

## Originality

The key idea in this paper: doing IRL in a way that estimates the expert's subjective beliefs is novel and promising.

## Clarity

The paper does a good job of setting up and proving the theoretical results.

## Significance

Depending on how useful these methods are in practical situations, this paper has the potential for high impact. It leverages a nice insight into IRL, supports that with solid theory, and demonstrates some initial empirical results.

# Weaknesses

## Quality

The experiments in this paper are not particularly strong. The results are primarily a comparison of two different ToM approaches, with less comparison of ToM approaches to other approaches. It is possible I have missed something with the experiments, but it was hard for me to identify the benefits of ToM in comparison to other IRL approaches (despite the claim in l.209). In either case, comparing to more baseline methods (clearly grounded in the related literature) would help.

Beyond this issue, the experiments are not super convincing as they are in simulation for a gridworld and Mujoco. I'm not sure how easy it would be to fix this, but a complex evaluation domain would help the paper.

Finally, a few related work pieces to include: https://arxiv.org/abs/1901.08654 estimates reward where the observed behavior has a subjective estimate of reward; https://dl.acm.org/doi/10.5555/3327345.3327381 deals with negotiating agents who have different dynamics estimates

## Clarity

It wasn't very clear how one could replicate the experiments or methods. Consider including pseudocode for the method and more details about, e.g., the baseline method. Furthermore, I found the analysis of the results (lines 254-258) incomplete. I recommend adding some more discussion of how and why performance differs across the conditions.

A few specific lines I found unclear:
  - l.116-117] Please say more about the distinction with (3);
  - [l.197] Please add more English description of the implications of the theorem, I had trouble parsing the math into an intuitive understanding

**Quality Of The Limitations Section:**

Limitations are addressed clearly

**Questions For Rebuttal:**

### Can you add a more complex robotics evaluation domain?

This would address my primary concern about the paper. I think the method is quite promising but I think that insufficient evaluation will reduce the paper's impact.

### Can you provide a more clear description of the method and baseline?

I think that more guidance for practitioners to implement the method will help the paper substantially. Similarly, in order to make the work more reproducible, it will help to have more descriptions of the baseline method for the experiment.

### Can you describe your changes to improve the clarity of the presentation?

Please mention the specific points I raised and any other changes you plan to make.

**Robotics Focus:**

Relevant but unlikely to deploy to hardware in near future

**Summary Of Paper:**

The paper proposes an approach to inverse reinforcement learning that accounts for the theory of mind of the demonstrator. In particular, this accounts for the demonstrator's subjective (and possibly incorrect) estimate of the dynamics. The authors analyze IRL in this framing and show how it has nice connections to robust learning from demonstrations. They provide experiments in a gridworld and on Mujoco benchmarks that demonstrate the approach.

**Summary Of Recommendation:**

I'm somewhat torn about this paper. I think the idea is quite promising, but the presentation and experiments have substantial room for improvement. With the paper as it is, I weakly recommend rejection. However, I think there are several points that can be addressed that improve my opinion of the paper.

Update after discussion/rebuttal: I’m happy with the authors response and changes. I still think that there is substantial room for improvement on the empirical work, but I value the combination of theoretical results and empirical validation. I’ve increased my score to recommend acceptance.

---

### Official Review · Reviewer_yDmY · 2023-07-21

**Confidence:** 3
**Originality:** Good
**Technical Quality:** Fair
**Clarity Of Presentation:** Fair
**Impact:** 3

**Recommendation:**

Weak Reject: I recommend rejecting the paper, but will not argue for my recommendation if the majority of other reviewers have a different opinion.

**Review:**

Strengths:
+ Empirical results show that the BTOM and RTOM approaches outperform an ML-IRL baseline.
+ Authors derive an approach that scales to high-dimensional tasks.

Weaknesses:
- I do not understand why the authors use a BTOM approach. There is no motivation for this approach and it does not seem to match the experimental domains.

- The authors do not assume that the expert has access to the true parameters of the environment, but all experiments assume this.
- Some of the derivations are hard to follow or not explained.

- The approach seems identical to other approaches except for the addition of the prior. However, as noted later the prior seems hard to justify in the setting of BTOM.

---
Post rebuttal:
I think the paper is stronger, and I think it is better framed by not using TOM wording (unless the authors were to address cases where the expert does not have a good model of the transition dynamics). However, I still do not see a significant win over ML-IRL and as such I'm not sure when this method would be chosen over existing methods. I think the paper needs to more clearly show why/when a Bayesian approach is better. Some ideas are to add experiments showing where ML-IRL has a significant drop in performance but the proposed Bayesian approach maintains high performance because of the robustness property. Other ideas include showing that the proposed approach is more data or computationally efficient. Prior work has shown that reward learning methods that seek point estimates are less robust than Bayesian robust counterparts (https://arxiv.org/pdf/2106.06499.pdf) and I would like to see something similar for this paper. Bayesian methods are also often useful for active learning and uncertainty quantification, but there is no evidence of this benefit in the current paper. I maintain my recommendation of weak reject.

**Quality Of The Limitations Section:**

Additional details required

**Questions For Rebuttal:**

Typos:
Line 61 R(s,a)
Eq (7) Should be \hat{P}, right?

Why use the prior in equation (6)? This is based on actual transitions and assumes that a dynamics parameter setting has higher likelihood if the actual transition is high under the model of the expert's dynamics. However, it is common that an expert can have biases and not understand the dynamics. In this case they would take an action expecting a different next state than the one that actual occurs. The current prior in the paper will assign this low likelihood even if this is the expert's true model of the env.


How do you get from (7) to (11)-(12)? Please add more explanation to the text.

Line 152. Why decompose a likelihood from prior work. I thought this would be decomposing Eq (7) from this paper.

In Equation (14) it is unclear what expectations are taken wrt the demonstrations.

Why use length 50 trajectories in a 5x5 grid world?

The related work section mentions several model-based offline IRL algorithms. Why not compare against them? Adding more baselines would strengthen the empirical results.

Isn't BTOM for when the demonstrator doesn't know the true dynamics. Please test in this setting and discuss why the BTOM mind approach is appropriate when the expert has perfect knowledge of the true dynamics.

**Robotics Focus:**

Relevant but unlikely to deploy to hardware in near future

**Summary Of Paper:**

This paper proposes a model-based offline IRL approach based on Bayesian theory of mind where the learner must estimate the demonstrator's reward function and dynamics. The authors show a connection to robust MDPs and demonstrate their algorithm on a grid world and on D4RL benchmarks.

**Summary Of Recommendation:**

The empirical results show an improvement; however, the general approach is poorly motivated and not studied in the setting most appropriate for BTOM. It is not clear why the approach works better than other approaches nor whether it will work in cases where the demonstrator does not have access to the true dynamics. The authors only compare against one baseline. Furthermore the mathematical details are often hard to follow.

---
Post rebuttal: see above notes in review. I think the paper needs more convincing results that there is value to the proposed approach as opposed to ML-IRL. Currently the gains seem mostly marginal and given the error bars it's unclear whether they are statistically significant. I maintain my recommendation of weak reject.

---

### Decision · Program_Chairs · 2023-08-30

**Decision:**

Accept (Poster)

**Comment:**

Reviewers appreciated the novelty of the proposed approach to IRL, the inclusion of theoretical results, and the thorough response in the rebuttal. The additional results with more baselines were very much appreciated, and were felt to make the paper stronger, although some doubts with regard to a clear performance win over ML-IRL remained (and should be addressed).